# The Implications of Cannabinoid-Induced Metabolic Dysregulation for Cellular Differentiation and Growth

**DOI:** 10.3390/ijms241311003

**Published:** 2023-07-02

**Authors:** Tina Podinić, Geoff Werstuck, Sandeep Raha

**Affiliations:** 1The Department of Pediatrics and the Graduate Program in Medical Sciences, McMaster University, Hamilton, ON L8S 4K1, Canada; podinict@mcmaster.ca; 2Department of Medicine and the Thrombosis and Atherosclerosis Research Institute, David Braley Research Institute, McMaster University, Hamilton, ON L8L 2X2, Canada; geoff.werstuck@taari.ca

**Keywords:** stem cells, cannabinoids, mitochondria, cellular differentiation, endoplasmic reticulum, oxidative stress, cellular metabolism

## Abstract

The endocannabinoid system (ECS) governs and coordinates several physiological processes through an integrated signaling network, which is responsible for inducing appropriate intracellular metabolic signaling cascades in response to (endo)cannabinoid stimulation. This intricate cellular system ensures the proper functioning of the immune, reproductive, and nervous systems and is involved in the regulation of appetite, memory, metabolism, and development. Cannabinoid receptors have been observed on both cellular and mitochondrial membranes in several tissues and are stimulated by various classes of cannabinoids, rendering the ECS highly versatile. In the context of growth and development, emerging evidence suggests a crucial role for the ECS in cellular growth and differentiation. Indeed, cannabinoids have the potential to disrupt key energy-sensing metabolic signaling pathways requiring mitochondrial-ER crosstalk, whose functioning is essential for successful cellular growth and differentiation. This review aims to explore the extent of cannabinoid-induced cellular dysregulation and its implications for cellular differentiation.

## 1. Endocannabinoid Signaling

The endocannabinoid system (ECS) is an endogenous system of neuromodulatory signaling that includes anandamide (AEA) and 2-arachidonoylglycerol (2-AG), collectively referred to as endocannabinoids, as well as the target receptors and enzymes that regulate endocannabinoid homeostasis. Canonical endocannabinoid receptors include cannabinoid receptor type 1 (CB1) and cannabinoid receptor type 2 (CB2), which are expressed at varying levels across different tissues [1,2]. The activation of CB1 and/or CB2 receptors leads to fast or slow downstream signaling events in the form of modulation of electrochemical gradients or induction of intracellular signaling cascades (e.g., cAMP-PKA, ERK, MAPK, PI3K), respectively. Synthetic endocannabinoids and phytocannabinoids have all been shown to interact with other receptors, including vanilloid receptors TRPv1, TRPv2, and TRPA1, G-protein coupled receptors (GPCRs) GPR55, GPR18, and GPR19, as well as nuclear receptors, such as the peroxisome proliferator-activated receptor gamma (PPARγ) [3,4].

Since initially being discovered in the brain, CB1 receptors have been well-characterized in the coordination of several brain processes, such as motor function, memory formation, cognition, and even appetite regulation [5,6]. Interestingly, at the cellular level, CB1 activation has been linked to adult neurogenesis in the subventricular zone (SVZ) in a murine neuronal model. Xapelli et al. showed that stimulating CB1R with the synthetic endocannabinoid (R)-(+)-Methanandamide (R-m-AEA) increased cellular proliferation after 48 h and promoted cellular differentiation of murine neural stem cells (NSCs) after 7 days [7]. CB2 receptors are more broadly expressed in peripheral tissues, such as immune cells, where they facilitate immunomodulation [8,9]. Moreover, CB2 agonists have been demonstrated to promote chemotaxis and enhance colony formation in hematopoietic stem cells (HSCs) [10]. Together, these key findings establish a crucial position for ECS signaling in tissue regeneration and development, which will be explored further below.

Although both CB1 and CB2 are classified as GPCRs embedded in the plasma membrane, these receptors are activated and antagonized by different classes of cannabinoids with different potencies [11]. For instance, endogenous AEA is a partial agonist of CB1 and a weak partial agonist of CB2 [12]. Likewise, cannabinoids derived from the cannabis plant, or "phytocannabinoids,” are able to interact with CB1 and/or CB2 receptors with varying affinities. A major cannabinoid derivative from the cannabis plant, delta-9-tetrahydrocannabinol (THC), has been shown to partially agonize both CB1 and CB2 receptors, whereas another major phytocannabinoid, cannabidiol (CBD), is a negative allosteric modulator of CB1 and a weak antagonist of CB2 [11].

## 2. Cellular Differentiation

Across embryonic development to late adulthood, highly regulated cellular differentiation is imperative for proper development and growth, as well as for the maintenance of specialized tissues throughout life. In general, this crucial cellular process underlies organogenesis and tissue regeneration, and its dysregulation or pathological dysfunction may accelerate aging and/or the onset of disease. Furthermore, the effects of cannabinoids on cellular differentiation are seen across a broad variety of tissues, including many peripheral tissues such as muscle, bone, and blood [13]. While we will not address the comparison of signaling pathways between tissues, our review will focus more specifically on central pathways related to mitochondrial function and endoplasmic reticulum (ER) stress, both of which are well known to impact cellular differentiation.

### 2.1. Stem Cell Characteristics

Cells that have the ability to develop into more specialized cell types by undergoing differentiation are collectively referred to as "stem cells." The dynamic range and ability of stem cells to differentiate into more mature cell types allow them to be further categorized in terms of cell potency [14]. For instance, stem cells that can potentially differentiate into all cell types are totipotent, whereas stem cells that can differentiate into most but not all cell types are pluripotent. Embryonic stem cells (ESCs) are defined as totipotent and pluripotent, as they give rise to multipotent stem cells. Additionally, other stem cells have the ability to differentiate into a specifically related family of cell types and are thus multipotent. For example, NSCs are among the most well-appreciated multipotent stem cells, with a large body of literature dedicated to adult neurogenesis that takes place within the SVZ and hippocampal regions [15]. Multipotent mesenchymal stem cells (MSCs) can differentiate into bone, cartilage, muscle, and skin tissues. Immune cell populations originate from a common multipotent hematopoietic stem cell [16]. 

All stem cells share two fundamental characteristics: their inherent ability to both (1) self-renew and (2) differentiate into specialized cell types [17]. To elaborate, undifferentiated or partially differentiated stem cells receive metabolic signals that induce them to either self-renew, thereby replenishing the existing stem cell population, or to commit and differentiate into specialized tissues through asymmetric division [18]. The tendency of stem cells to favor one fate decision over another is dependent on several intracellular and extracellular signaling cues. Stem cells are not constitutively active; rather, a quality control mechanism exists wherein a subset of stem cells enter quiescence, a state of metabolic depression, to maintain a functional pool of stem cells throughout adulthood [19]. Within this subset of metabolically inactive stem cells, there exist naïve and primed pluripotency states, which possess inherent epigenetic distinctions [20]. In humans, the former closely resemble premature embryonic epiblast stem cells, and the latter resemble post-implantation epiblast cells [20,21]. The dynamic ability of stem cells to maintain or exit pluripotency is reliant on the coordination of pro-differentiation or pro-self-renewal genes at the transcriptional level.

### 2.2. Key Intracellular Signaling Pathways Involved in Stem Cell Function and Differentiation

In response to chemical signals in their microenvironment, stem cells undergo transcriptional changes at the nuclear level to facilitate developmental and regenerative processes. Extrinsic pressures due to oxygen tension and the presence of inflammatory cytokines have been demonstrated to mediate stem cell differentiation by altering cellular transcriptional programs [22]. Stem cells are maintained through the actions of the Wnt and Notch signaling pathways, both of which have been implicated in promoting “stemness” [23,24,25,26,27,28]. In general, the canonical Wnt/β-catenin signaling pathway leads to the stabilization of β-catenin, followed by the accumulation and localization of β-catenin in the nucleus, where it crucially regulates genes related to pluripotency [29,30]. In naïve human embryonic stem cells (hESCs), Xu et al. demonstrated that the inhibition of Wnt signaling decreases the proliferative capacity of the naïve stem cell population and further promotes a primed stem cell phenotype [31]. The addition of recombinant Wnt3 partially rescued the proliferative potential of hESCs, demonstrating that Wnt signaling is critical in maintaining self-renewal [31]. During placentation in early pregnancy, hypoxic conditions (2.5% O_2_) dominate the trophoblast microenvironment before the reestablishment of normoxic levels (20% O_2_) following placentation [32]. 

In some cases, stem cells respond to oxygen deprivation by recruiting stable hypoxia-inducible factors (HIFs), which promote vascularization and preserve cellular homeostasis through the regulation of bioenergetic pathways. Activated HIF transcription factor subunits localize to the nucleus and bind to the hypoxia response element (HRE), which promotes the transcription of cell proliferation and survival genes. However, the transcription factor HIF2α has also been shown to activate Wnt and Notch signaling pathways. To illustrate the key role of HIFs during trophoblast stem cell differentiation, an early study found that HIF1α- and HIF2α- null embryos fail to undergo placental morphogenesis as a result of dysfunctional stem cell fate determination [33]. Interestingly, Caniggia et al. observed substantial levels of HIF1α and TGFβ3, an inhibitor of extravillous trophoblast (EVT) differentiation, during early pregnancy [34], and further uncovered that HIF1α is located upstream of TGFβ3 gene expression, thus mediating cell fate genes [35].

The exact role of inflammation in mediating stem cell homeostasis is still unclear [36]. Some studies have reported a negative regulatory role for inflammation, wherein stem cell proliferation is diminished [37]. To emphasize the inhibitory role of inflammation on neurogenesis, Monje and colleagues found that treatment with indomethacin, an anti-inflammatory drug, rescued neural stem cell function following endotoxin-induced inflammation [38]. Likewise, in a mouse model of cortical development, it was observed that induced, systemic maternal inflammation resulted in less ventricular proliferation in the fetus, indicating negative regulation of stem cell function [39]. Finally, in mice lacking the tumor necrosis factor receptor 1 (TNFR1), it was shown that cell proliferation was considerably elevated in the dentate gyrus, suggesting that negative regulation might be the result of pro-inflammatory responses mediated by TNFR1 [37]. In contrast, Wolf et al. demonstrated that neural stem/progenitor cells (NSPCs) in the mouse hippocampus display enhanced proliferative capacity following bacterial endotoxin-induced inflammation [40]. Previous in vitro studies have identified key inflammatory cytokines, TNFα and IL-1β, as promoters of NSPC proliferation and differentiation by activating either NFKβ or JNK signaling pathways [41,42]. Taken together, it is apparent that inflammation plays a complex and multifaceted role in mediating stem cell function.

In addition to oxygen-sensing mechanisms, there exist various nutrient-sensing systems designed to coordinate cellular homeostasis with nutrient availability. Among these, the mammalian target of rapamycin (mTOR) protein kinase and AMP-activated kinase (AMPK) emerge as fundamental intracellular nutrient sensors with opposing downstream effects [43]. To elaborate, the activation of mTOR enhances anabolic processes, including cell growth, protein translation, and mitochondrial metabolism, whereas, under low nutrient conditions, AMPK activation promotes catabolic activities, such as glucose metabolism and autophagy. The cross-regulation of autophagy by mTOR and AMPK points to a crucial role in autophagic processes in stem cell dynamics. In HSCs and satellite cells, the deletion of an autophagy-related gene, Atg7, resulted in the loss of both HSC and satellite cell pools, phenotypes that were associated with increased ROS production and the accumulation of damaged mitochondria [44,45]. Likewise, Atg12-deficient HSCs exhibited similar deficits in HSC self-renewal [46], suggesting that autophagy might play a role in removing mitochondria to regulate stem cell bioenergetics. mTORC1 is complexed with several proteins, including the regulatory-associated protein of mTOR (Raptor) and the DEP-domain-containing mTOR interacting protein (Deptor), which have a range of functions from substrate recognition to the regulation of mTOR activity, respectively. The major downstream targets of mTOR, a serine/threonine kinase, include the ribosomal protein S6 kinase (S6K1) and the eukaryotic initiation factor 4E-binding protein (4EBP1), which function in amino acid and ATP sensing. Specifically, being fundamental components for protein synthesis, amino acids indeed promote mTORC1 complex assembly, while ATP availability supports the energy requirement for downstream anabolic processes [47]. Upon its phosphorylation and activation by mTORC1, pS6K1 further phosphorylates eIF4B, a cofactor required for mRNA translation initiation due to its RNA helicase activity, and PDCD4, an inhibitor of eIF4B, thus promoting its degradation [48]. Alternatively, the native function of 4EBP in preventing the formation of the eIF4E complex required for translation by binding to and inhibiting eIF4E is hindered by mTORC1 phosphorylation, which ultimately dissociates 4EBP from eIF4E [49]. In addition, these mTORC1 effectors may possess multifaceted roles in cellular growth. For instance, de novo lipogenesis can be induced directly through an mTORC1-mediated pathway, wherein S6K1 activates the sterol-responsive element binding protein (SREBP), or indirectly, in which the absence of mTORC1 signaling leads to the association of Lipin1 and SREBP, thus preventing its activation [50]. Taken together, a complex interplay of feedback loops guiding mTORC1 signaling is responsible for promoting anabolic processes during conditions of substrate abundance. 

In the context of aging, it is becoming increasingly apparent that mTOR contributes to tissue homeostasis by modulating stem cell maintenance, differentiation, and proliferation [51]. In a phosphoinositide 3-kinase (PI3K)/Akt-dependent manner, mTORC1 has been shown to promote the differentiation or proliferation of NSCs, HSCs, and mammary and germline stem cells upon stimulation by growth factors such as IGF [52,53,54]. IGF-mediated stimulation has been implicated in neuronal differentiation, wherein it induces an intracellular signaling cascade that results in the phosphorylation of Akt at Ser473 and Thr308 residues in differentiating olfactory bulb stem cells (OBSCs) [55]. Likewise, exit from pluripotency and initiation of differentiation have been linked to mTOR in hESCs, where these signaling cascades are tightly regulated. Easley et al. knocked down the rapamycin-insensitive companion of the mammalian target of rapamycin (Rictor), an associated protein of the mTORC2 complex, and tuberous sclerosis complex 2 (TSC2), an inhibitor of mTORC1 [56], using siRNA-mediated technology, and found increased activation of p70 S6K coupled with greater differentiation in hESCs [57]. In contrast, a recent study by Lee et al. found that inhibiting the well-established negative regulator of PI3K signaling, phosphatase, and tension homolog (PTEN), led to increased human NSC proliferation [58]. Furthermore, Schaub et al. demonstrated that mTORC1 and mTORC2 are involved in coordinating osteoblastic differentiation in MSCs [59]. In brief, MSC differentiation is achieved in vitro by incubating cells in an osteoblast induction medium for three weeks, after which the expression of key osteoblast marker proteins, such as osteopontin, collagen I and III, and Cbfa1, is highly expressed. However, it was found that osteoblast marker proteins were markedly reduced in MSCs treated with rapamycin, an mTOR inhibitor [59]. More specifically, their findings revealed that rapamycin exposure led to decreased phosphorylation of p70-S6K, a downstream effector of mTORC1, whereas mTORC2 activity was increased under the same conditions. Taken together, these observations suggest the existence of an intricate feedback loop between intracellular signaling targets, mTORC1, and mTORC2, in their co-regulation of cellular function.

## 3. Metabolic Regulation of Stem Cell Fate Decisions

### 3.1. Mitochondrial Dynamics

Emerging evidence emphasizes the critical regulation of cell fate programmes through intricate metabolic networks coupled with intracellular cross-talk between organelles. To begin, the role of mitochondrial dynamics and metabolism in regulating cell fate decisions has been well appreciated, considering that mitochondrial morphological alterations directly influence the metabolic landscape. Mitochondrial dynamics refers to the inherent ability of mitochondria to undergo fusion or fission processes in response to microenvironmental conditions. Elongated mitochondria preferentially promote oxidative phosphorylation (OXPHOS), often characterized as an oxygen-rich, high-ATP-producing energy state, due to their well-defined and abundant cristae folds that provide greater surface area to accommodate electron transport chain (ETC) machinery. Since mitochondria are double-membrane bound, mitochondrial fusion is accomplished through the combined activities of dynamin-related GTPases, optic atrophy protein 1 (Opa1), and mitofusins 1 and 2 (Mfn1/2), which govern inner mitochondrial membrane (IMM) fusion and outer mitochondrial membrane (OMM) fusion, respectively [60,61]. Additionally, the ultrastructure of mitochondria and the extent of cristae remodeling are heavily contingent upon Opa1 activity, a mechanism through which cytochrome *c* release is tightly controlled, further suggesting a critical role for mitochondrial dynamics in programmed cell death. [62].

Alternatively, mitochondrial fragmentation is linked to a glycolytic energy state and contributes to quality control (QC) mechanisms within the cell by providing additional mitochondria to support cell growth and division while priming damaged mitochondria for mitophagy [63,64]. Mitochondrial fission occurs when a cytosolic form of dynamin-related protein 1 (Drp1) is phosphorylated on specific serine residues, such as serine 616 (S616), by active ERK2 during MAPK signal transduction, thus promoting its subsequent translocation to the mitochondrial membrane [65]. Here, Drp1 aggregates with either mitochondrial fission 1 (Fis1) protein or mitochondrial fission factor (MFF) to facilitate cleavage of the mitochondrion [66]. In contrast, the phosphorylation of serine 637 (S637) by PKA terminates Drp1 activity and promotes mitochondrial elongation [67]. Evidently, post-translational modifications to the phosphorylation status of key mitochondrial dynamics markers can critically regulate their activity.

### 3.2. Metabolic Switch as a Driver of Stem Cell Differentiation

As mentioned above, the mitochondrial phenotype is dominated by either OXPHOS or aerobic glycolysis, both of which generate ATP and, to some degree, metabolic by-products such as mitochondrial reactive oxygen species (mtROS). Contrary to the popular narrative that free radicals compromise cellular integrity, increasing evidence is pointing to a crucial cellular signaling role for mtROS [68,69]. For instance, following the stimulation of rat vascular smooth muscle cells (VSMCs) by platelet-derived growth factor (PDGF), Sundaresan et al. observed elevated levels of hydrogen peroxide (H_2_O_2_) [70]. Interestingly, these PDGF-induced cellular effects were inhibited once H_2_O_2_ was neutralized, suggesting that free radicals could participate in growth signaling [70]. In general, mtROS production is relatively low during aerobic glycolysis and becomes elevated during OXPHOS, with the majority of mtROS being generated from electron leakage that occurs at Complexes I and III [71,72].

A growing body of evidence suggests a crucial role for mitochondria upstream of stem cell fate decisions. Generally, self-renewing stem cells are primarily reliant on aerobic glycolysis and characterized by low levels of intracellular ROS. Aerobic glycolysis is favorable for self-renewing cells as it provides both metabolic intermediates and increased rates of ATP production to support cell division [73]. At the onset of stem cell commitment and differentiation, stem cells undergo an apparent metabolic shift from glycolysis to OXPHOS, during which ROS and ATP production are increased to sustain escalating ATP demands [74,75,76,77]. Indeed, these transient metabolic changes are accompanied by mitochondrial morphological changes in stem cell populations [78]. Thus, prior to and during cellular differentiation, predominantly fragmented and spherical mitochondria in primitive stem cell populations emerge as elongated mitochondria containing well-defined cristae, capable of supporting oxidative respiration [79,80,81]. To further emphasize the link between mitochondrial morphology and stem cell fate decisions, recent literature has demonstrated that mitochondrial dynamics occur upstream of stem cell fate decisions. In NSCs, the deletion of mitochondrial fusion proteins, Opa1 and Mfn1/2, promoted mitochondrial fragmentation and neural stem cell commitment and differentiation, suggesting that acute metabolic shifts confer stem cell functionality. Further, they elucidated a role for ROS in activating an NRF2-dependent pathway that suppressed self-renewal genes and promoted the expression of differentiation genes [82]. Similarly, Luchsinger and colleagues observe that although quiescent long-term hematopoietic stem cells (LT-HSC) possess primarily fragmented mitochondria, their differentiation into short-term hematopoietic stem cells (ST-HSC) is coupled with elevated levels of Mfn2 and mitochondrial elongation [83]. In addition, this differentiation is seemingly impaired following the deletion of Mfn2, as the LT-HSC pool remains consistent in size [83]. Overall, an overwhelming body of literature recognizes the integrated role of mitochondrial dynamics during stem cell differentiation (Figure 1).

### 3.3. Endocannabinoid Signaling at Mitochondrial CB1 (mtCB1)

Previous studies have mainly focused on elucidating the effects of THC on cellular functions such as apoptosis, respiration, and metabolism. Jia et al. conducted in vitro drug treatments with 10 μM of THC in Jurkat cells and found that THC suppresses downstream MAPK effectors and leads to the translocation of a pro-apoptotic Bcl-2 protein, Bad, to mitochondria where apoptosis is initiated [84]. Treatment of BeWo placental cells with 20 μM THC was associated with elevated mtROS production and oxidative stress, decreased ATP production, increased mitochondrial fragmentation as evidenced by elevated Drp1 expression, and reduced mitochondrial membrane potential after 48 h [85,86]. In isolated mitochondria, 30 µM of CBD induces mitochondrial calcium (Ca^2+^) overload and subsequent formation of mitochondrial permeability transition pores (mPTP), causing mitochondrial swelling due to an influx of permeable small molecules and eventually cytochrome *c* release [87]. Loss of cytochrome *c* from the mitochondrial matrix disrupts electron transport chain (ETC) function, which is evident by the apparent loss of ATP production, and is further involved in apoptosome formation in the cytoplasm. It has been postulated that CBD may exert these mitochondrial effects by interacting with the mitochondrial voltage-dependent anion channel 1 (VDAC1), thereby promoting a configurational change to a “closed” state that attracts Ca^2+^ accumulation in the intermembrane space. Interestingly, <10 µM of CBD promoted autophagy and cell death, whereas ~1 µM CBD seemingly stimulated cell proliferation, which suggests that Ca^2+^ regulation by cannabinoids is mediated through mitochondria and may depend on the doses employed [87]. In addition, evidence from human bronchial epithelial cells suggests concentration-dependent effects of THC, and a synthetic analogue, CP55,940, induced cellular Ca^2+^ fluxes via both store-dependent capacitive Ca^2+^ entry routes as well as non-capacitive routes [13,88]. While a detailed discussion of these routes is beyond the scope of this review, the possibility that cannabinoids can impact complex Ca^2+^ signaling pathways that impact cellular differentiation and function should be borne in mind. Gross et al. also observed a dose-dependent decrease in mitochondrial oxygen consumption in CBD-treated humans and canine gliomas, furthering the idea that cannabinoids may interact with and dysregulate mitochondrial function [89]. 

Until recently, the effects of THC and CBD on cellular function were mainly attributed to their activity at the plasma membrane-bound cannabinoid receptors; however, this notion has been challenged following the discovery of a novel mitochondrial CB1 (mtCB1) receptor [90]. Specifically, work by Bénard et al. identified drastic differences in the expression of mtCB1 receptors in wild-type compared to CB1 knockout (CB1-KO) mice, noting that 30% of CB1+/+ CA1 mitochondria expressed mtCB1 [90]. Additional investigations revealed that THC directly activates mtCB1, triggering an intramitochondrial G-protein signaling cascade wherein Complex I is destabilized and mitochondrial respiration is diminished [91]. The genetic sequence encoding mitochondrial CB1 was elucidated by Hebert-Chatelain and colleagues, who identified that the first 22 N-terminal amino acids in the coding region of CB1 account for its mitochondrial localization [92]. Using primary murine hippocampal cells, Ma et al. investigated how mtCB1 regulates mitochondrial function following cerebral ischemia/reperfusion (I/R) injuries and discovered that ACEA-induced mtCB1 receptor activation diminished Ca^2+-^induced mitochondrial damage, which was fully blocked and only partially blocked by cell-permeant or cell-impermeant CB1 receptor antagonists, respectively [93]. Similarly, mtCB1 was identified by electron microscopy in hypothalamic pro-opiomelanocortin (POMC) neurons, where it may function by facilitating mitochondrial adaptations required for the activation of these neurons, thus stimulating the sensation of hunger [94]. Importantly, recent studies have identified mtCB1 in male and female reproductive tissues. The distinct localization of mtCB1 to murine ovarian interstitial glands, which are primarily progesterone-producing cells, suggests a potential implication for this novel receptor in steroidogenesis [95]. This finding is consistent with an earlier discovery of mtCB1 in sperm cells. Here, Aquila et al. determined that AEA, an mtCB1 agonist, plays a role in sperm survival and facilitates the acrosome reaction [96]. Finally, mtCB1 has also been found in muscle tissue, where its known function is currently limited to regulating mitochondrial respiration [97]. Therefore, although the role of mtCB1 remains widely unrecognized in the broader context of cellular function, it appears that cannabinoid signaling influences mitochondrial dynamics and respiration, regardless of whether these changes are mediated through the mitochondrial or plasma membrane-bound receptor.

## 4. Endoplasmic Reticulum (ER) and the ER Stress Response

### 4.1. Structure of the ER

Aside from being the largest membrane-bound organelle, the endoplasmic reticulum (ER) has been primarily recognized for its roles in protein synthesis, calcium homeostasis, and lipid biosynthesis. In order to develop an appreciation for these crucial cellular roles, we must first emphasize some key structural domains. To begin, the ER periphery is comprised of membrane projections referred to as either cisternae (“sheets”) or tubules. While the cisternae are densely covered in ribosomes, which are the cellular machinery that carries out protein translation and assist in colocalization to the ER lumen, the tubules, on the other hand, extend as finger-like projections towards the cytosol and have relatively fewer ribosomal units [98]. As such, with reference to their relative ribosomal content, cisternae and tubules are referred to as "rough" ER and "smooth" ER, respectively. Importantly, the branching nature of ER tubules implies their contact with other intracellular organelles, which are collectively referred to as membrane contact sites (MCSs), further implying that the ER contributes to highly coordinated cellular processes that require intracellular communication between organelles [98]. In fact, up to 20% of the mitochondrial surfaces are closely affiliated with ER membranes, and these contact sites are more specifically classified as mitochondria-associated membranes (MAMs) [99]. For instance, in a study conducted by Gomez-Suaga et al., they demonstrate that autophagy is closely regulated by mitochondrial-ER tethering between a mitochondrial protein, PTPIP51, and an ER-membrane-associated protein, VABP, wherein the depletion or overexpression of these proteins leads to increased or decreased autophagosome formation, respectively [100].

### 4.2. Function of ER in Key Cellular Processes

Although protein synthesis is highly coordinated, dysfunctional and misfolded proteins may arise. As a result, there are inherent protein quality control mechanisms in place, such as the ER-associated degradation pathway (ERAD), which provides a mechanism through which irregular polypeptides may be degraded [101]. Early studies first identified this lysosome-independent, ER-specific proteolytic pathway upon observing that T-cell receptor subunits could be degraded without the activity of lysosomal proteases [102]. Since then, the role of the ER proteolytic pathway, now commonly known as the ubiquitin-proteasome system (UPS), has been well characterized. In brief, proteins destined for degradation are tagged with special polyubiquitin chains that are recognized and cleared by the proteasome [103,104]. When protein degradation pathways fail and/or misfolded or abnormal proteins aggregate, the ER stress response pathway can become activated, which is prevalent in the progression of pathological conditions such as Alzheimer’s disease [105,106]. 

The ER membrane contains many distinct channels and receptors responsible for transporting Ca^2+^ across the membrane. Given that Ca^2+^ molecules act as secondary messengers in signal transduction pathways of all cell types and contribute to the broader physiological role of certain tissues, such as in the context of muscle contraction or neuronal firing, there is a critical need to regulate intracellular Ca^2+^ levels [107]. Normally, the levels of cytosolic Ca^2+^ are considerably lower (~100 nM) relative to those of the ER lumen (~100–800), indicating that the latter plays a role in calcium deposition [108]. In general, Ca^2+^ release is mediated by the activation of a GPCR that proceeds to stimulate PLC, after which activated PLC continues to facilitate the cleavage of phosphatidylinositol 4,5 bisphosphate (PIP2) into diacyl-glycerol (DAG) and inositol 1,4,5 triphosphate (IP3). The ER membrane contains both Ca^2+-^uptake channel sarco-endoplasmic reticulum Ca^2+^-ATPase (SERCA) pumps and several Ca^2+^-releasing channels, such as ryanodine (RyRs) and IP3 receptors (IP3R) [109]. In response to increased cytosolic calcium, SERCA pumps increase Ca^2+^ uptake into the ER lumen to prevent mitochondrial calcium overload, a signaling event that is otherwise present in most pro-apoptotic signaling cascades. In muscle tissue, the sarcoplasmic reticulum (SR) is reminiscent of the smooth ER, and once Ca^2+^ enters via SERCA pumps, it can be further stored by binding to calesequestrin [110]. Similarly, RyRs are found in several cell types, including neurons and epithelial cells, and they release calcium through a mechanism called Ca^2+^-induced Ca^2+^ release (CICR) upon sensing increased levels of calcium in the cytoplasm [111]. Alternatively, IP3R binds IP3 following the upstream signaling cascade induced by a substrate, wherein PIP2 is cleaved into sub-components, allowing Ca^2+^ to exit the ER lumen into the cytoplasm. Although Ca^2+^ transport is regulated by similar integral proteins and channels found in the ER, it is simultaneously contained within subcellular compartments encased by ER-mitochondria contact points. In this manner, unwanted Ca^2+^ leakage can be tightly controlled, given that mitochondria also possess voltage-dependent anion channels (VDACs) and mitochondrial calcium uniporters (MCU) that allow for the passage of Ca^2+^ from the microenvironment [112,113]. Aberrant Ca^2+^ accumulation reduces the efficiency of ER-resident protein folding chaperones, resulting in abnormal protein folding and concurrent activation of the unfolded protein response (UPR), which we will discuss in more detail later in the review.

Furthermore, the ER directly fosters cellular growth and differentiation by promoting lipid biosynthesis, through which it crucially provides structural support in the form of membrane-bound organelles and cellular membranes, as well as key signaling molecules and a greater capacity for energy storage. To emphasize this crucial interplay between lipid homeostasis and cellular function, human HSCs possess distinct lipid arrangements, particularly in their expression of an ER-membrane-embedded sphingolipid enzyme, DEGS1, which confers functionality to HSCs [114]. Ultimately, DEGS1 converts dihydroceramide to ceramide, after which ceramide is further metabolized into another bioactive lipid, sphingosine-1-phosphate (S1P). Interestingly, the modulation of S1P has been implicated in the cellular proliferation, differentiation, migration, and survival of MSCs, ESCs, HSCs, and immune cells [114,115,116,117]. The ablation or inhibition of DEGS1 induces autophagy [118], which promotes the maintenance of stemness programs [114].

### 4.3. ER and Mitochondrial Cross-Talk Mechanisms

The coordinated signaling processes that exist between mitochondria and ER are likely dependent upon the abundance of contact sites [119]. To re-emphasize, mitochondrial fission and fusion are crucial for cellular signaling, especially when it comes to mediating stem cell differentiation [120]. Notably, it has been demonstrated that the outer mitochondrial fusion proteins dually function in ER-mitochondria tethering, following the observation that Mfn1-Mfn2 multimers closely link these two organelles [121] In the absence of Mfn2 expression, these contacts are considerably weakened, making the mitochondria less sensitive to intracellular calcium levels [121,122]. Moreover, Friedman et al. used electron microscopy to visualize the extent of ER-mitochondria associations and revealed that ER tubules wrap around the mitochondrial outer membrane, ultimately leading to the decreased mitochondrial diameter at these sites. Essentially, they found that the ER engages in the process of mitochondrial fragmentation by priming the site of constriction prior to the recruitment and aggregation of Drp1 at the mitochondrial membrane, where division will take place, by tracking the translocation of GFP-tagged Drp1 to MAMs [119]. Alternatively, it was observed that the establishment of MAMs precedes Drp1 translocation and subsequent activity, as Drp1 knockdown resulted in no changes to these contact sites [119].

In addition, it has become increasingly apparent that mitochondrial morphology and ER morphology influence one another. In fact, Pitts and colleagues were among the first to discover that a dynamic-like protein, Dlp1, in yeast, which partially resembles mammalian Drp1, maintains mitochondrial morphology as well as ER morphology [123]. Furthermore, mitochondria-ER contacts are involved in mediating cellular processes, such as apoptosis, as they create microenvironments where Ca^2+^ ions may localize and accumulate (see Figure 2). Iwasawa et al. revealed that Fis1, a mitochondrial fission protein found in both mammalian and yeast cells, associates with the B-cell receptor-associated protein 31 (Bap31) linked to the peripheral ER membrane. They demonstrated that Fis1 promotes apoptosis by facilitating the cleavage of Bap31 into its pro-apoptotic form, p20Bap31 [124]. Additionally, they provide evidence for the coupling of pro-caspase 8 to the Fis1-Bap31 complex during the early phases of apoptosis initiation. Following these early stages of activation of apoptosis, the signaling cascade results in the release of ER calcium ions, which perpetuate apoptotic signaling in neighboring mitochondria [124]. Considering that mitochondrial VDAC1 is Ca^2+^-permeable, one study showed that mitochondrial Ca^2+^ overload is associated with VDAC1 oligomerization, a potential hallmark associated with apoptosis [125]. Once Ca^2+^ accumulates in the intermembrane space, mPTP formation occurs, and mitochondria become permeable to small molecules. Interestingly, Drp1 recruitment to the outer mitochondrial membrane [126] and its subsequent activation are also stimulated by Ca^2+^ effluxes from the ER lumen into the mitochondrial microenvironment, ultimately resulting in the release of cytochrome *c* into the cytoplasm and apoptosis induction. Despite studies supporting the role of Drp1 in apoptosis initiation, whether Drp1 is pro-apoptotic or anti-apoptotic remains debated in the literature [127].

### 4.4. ER-Dependent Unfolded Protein Response 

Prolonged endoplasmic reticulum dysfunction brought on by the accumulation of misfolded proteins and/or disruptions in Ca^2+^ gradients may induce the Unfolded Protein Response. Essentially, the UPR pathway implements changes at both transcriptional and translational levels in order to support a comprehensive response to ER stress. In general, there are three separate ER transmembrane proteins capable of sensing unfolded proteins and which activate distinct branches of the UPR: (1) inositol requiring enzyme 1α/β (IRE1), (2) PKR-like ER kinase (PERK), and (3) activating transcription factor 6α/β (ATF6) [128]. IRE1 contains two domains, an ER luminal domain and a cytosolic domain, which are responsible for sensing unfolded proteins and initiating UPR through its kinase activity [129,130]. Upon stimulation, IRE1 functions as an endonuclease and splices X-box-binding protein (XBP1) mRNA into a more potent splice variant, or XBP1s [131,132]. XBP1s is a transcription factor that increases protein folding activity and protein degradation, in order to lessen the pressure of misfolded and aggregated proteins in the ER lumen [133].

Upon activation, PERK-mediated kinase activity leads to the phosphorylation of eukaryotic translation initiation factor 2α (eIF2α), which ultimately inhibits ribosomal activity and briefly halts global translation [129,134]. The fundamental goal of PERK activation is to prevent additional protein translation to alleviate the protein folding burden on the ER. Aside from temporarily pausing protein translation, PERK-mediated signaling leads to the upregulation of activating transcription factor 4 (ATF4), which functions to increase antioxidant scavenging activity, thus alleviating ER stress [135]. Activated ATF6 translocates to the Golgi apparatus, where it can be cleaved by S1P and S2P proteases, thus releasing its cytosolic domain. This cytosolic portion, or bZIP transcription factor, enters the nucleus and upregulates UPR-related genes, such as ER chaperones, which are responsible for folding proteins [136]. Thus, all three branches are dedicated to mitigating ER stress in order to prevent further cellular catastrophes, such as the induction of apoptosis.

## 5. Endocannabinoid Signaling and Intracellular Functions

Despite the fact that phytocannabinoids (THC, CBD) share similar molecular targets as endocannabinoids (AEA, 2-AG), their individual activities ultimately rely on their distinct allosteric regulation at each cannabinoid receptor, suggesting that these classes of cannabinoids should be examined independently to better understand their effects. Earlier we emphasized how mitochondria possess certain structural features, such as the mtCB1 receptor, which impart an elusive role for the energy-producing organelle in the context of endocannabinoid signaling. In addition, it is clear that mitochondria and ER are highly interconnected and that both organelles contribute to several cellular functions, such as apoptosis. As such, it is crucial to explore the way endocannabinoids affect mitochondrial and ER function and signaling.

### 5.1. Regulation of Mitochondrial Function through Endocannabinoid Signaling

In order to develop a comprehensive understanding of mitochondrial function, several parameters pertaining to mitochondrial respiration and the oxidative state must be considered, including mitochondrial oxygen consumption rate (OCR), mitochondrial dynamics, mitochondrial membrane potential, ATP production, the extent of oxidative stress (or mtROS production), and mitochondrial Ca^2+^ regulation. An early study revealed that, compared to tobacco, cannabis smoke exposure disrupted mitochondrial bioenergetics in epithelial cells [137]. In particular, they demonstrated that rats exposed to cannabis smoke showed a 75% reduction in ATP production and decreased red fluorescence of a cationic carbocyanine dye, JC-1, indicating diminished mitochondrial membrane potential [137,138] (Figure 3). In establishing that mitochondria are implicated in endocannabinoid signaling, the more recent literature has focused on elucidating the roles of cannabinoid receptor agonists, AEA and 2-AG, in mediating mitochondrial functions. Athanasiou and colleagues were among the first to characterize the impacts of AEA in vitro, observing that 100 mM AEA induced an apoptotic phenotype in human lung cancer cells (H460 cell line). Specifically, they noted an increased appearance of rounding and unadhered cells combined with increased cytoplasmic granularity following AEA treatment [139]. Like other cannabinoid receptor agonists, such as THC, they demonstrated that AEA treatment in isolated rat mitochondria led to decreased mitochondrial membrane potential upon quantifying Rhodamine 123 fluorescence using fluorimeter tracing [139]. Most notably, it was discovered that incremental increases in AEA concentration resulted in slightly enhanced complex II-III activity at low micromolar doses, which was short-lived as complex II-III activity drastically decreased while approaching the higher micromolar range [139]. This evident disruption in ETC activity may be attributed to mtCB1 activation, as Bénard et al. have previously shown that mtCB1 signaling results in decreased PKA-dependent phosphorylation of NDUFS2, a crucial complex I subunit, thus decreasing mitochondrial respiration [90]. In addition, work by Catanzaro et al. suggests that 50 µM of AEA operates in a time- and dose-dependent manner to increase mitochondrial swelling and that pretreating simultaneously desensitizes mitochondria to intracellular Ca^2+^ levels by decreasing cytochrome *c* release in response to calcium [140]. As mentioned earlier, mPTP allows for small molecules to enter mitochondria and ultimately leads to mitochondrial swelling as a result of Ca^2+^ influx and ultimately cytochrome *c* release. One study exhibited that cyclosporin A (CsA), an mPTP inhibitor, prevented Ca^2+^-induced cytochrome *c* release and showed that 2-AG inhibits CsA-sensitive Ca^2+^-dependent cytochrome *c* release in rat liver mitochondria [141]. Overall, given that endocannabinoids possess distinct spatial and temporal distribution patterns in various physiological systems, including reproductive, neuronal, and musculoskeletal systems, in order to drive intracellular processes and/or maintain tissue homeostasis, their effects on mitochondrial function are worth investigating further.

### 5.2. Regulation of ER Function through Endocannabinoid Signaling

Endocannabinoids may influence the delicate balance that exists between pro-survival and pro-apoptotic pathways mediated by ER function. Several studies have identified AEA as a potent inducer of apoptosis in tumor cells; however, the signaling pathway(s) through which AEA may be promoting apoptosis are not fully understood. A proteomic analysis of AEA-treated cells revealed an upregulation of several proteins, notably BiP (GRP78), in parallel with increased apoptosis. GRP78 is an ER molecular chaperone with several crucial functions, such as modulating the UPR by inhibiting the major UPR receptors: PERK, IRE1, and ATF6 [142]. Interestingly, Pasquariello and colleagues found that treatment with SR141716, a CB1 antagonist, reduced the expression of GRP78 and apoptosis in the presence of AEA, suggesting that apoptosis induction is regulated through CB1 activity. Additionally, considering that AEA stimulation of CB1 involves p38 and p42/44 MAPK downstream, the inhibition of these signaling factors eliminated AEA-induced apoptosis, which further confirmed that CB1 signaling was involved [143]. Cyclooxygenase-2 (COX2) is an enzyme responsible for metabolizing AEA and is primarily localized in the ER and nuclear regions, thus positioning AEA metabolism in the ER. Using tumorigenic keratinocytes, Soliman et al. found that AEA activated all three major pathways of the UPR (PERK, IRE1, and ATF6). In addition, AEA activated and upregulated ER stress response proteins, including C/EBP homologous protein-10 (CHOP) and cleaved caspase-3, which are well-known indicators of ER stress and apoptosis. Furthermore, these effects were neutralized upon inhibiting ER stress, indicating that the pro-apoptotic signaling cascade initiated by AEA is indeed ER-stress dependent [144]. Additionally, they showed that AEA cytotoxicity is partially reliant on its ability to increase oxidative stress due to the fact that pre-treatment with the antioxidant N-acetyl cysteine (NAC) inhibited AEA-induced oxidative stress and apoptosis. Considering that pre-treatments with either AM251 (a CB1 antagonist) or AM630 (a CB2 antagonist) did not alter the AEA-induced caspase 3 cleavage and consequent cell death, it was concluded that AEA pro-apoptotic activity is cannabinoid-receptor independent and instead depends on ER stress response activation [145]. Alternatively, another major endocannabinoid, 2-AG, has been implicated in trophoblast cell turnover by promoting ER-stress-induced apoptosis [146]. Almada et al. discovered that upon treating BeWo cells with 10 uM 2-AG, there was increased GRP78 gene expression, an ER-resident chaperone in charge of mediating ER stress signaling pathways, indicating the initiation of the UPR response (Figure 3). Additionally, they showed that 2-AG was able to increase the expression of ATF4, an upstream transcription factor responsible for CHOP activation and subsequent cell death, whereafter CB1 receptor antagonism failed to rescue the phenotype and CB2 receptor antagonism reversed the effects of 2-AG [146]. Moreover, another study attributed the pro-apoptotic activity of cannabinoids to the upregulation of the transcriptional co-activator p8 and its transcriptional target, pseudo-kinase tribble homolog 3 (TRB3) [147]. To emphasize this relationship, Salazar et al. found that the well-known phytocannabinoid and cannabinoid receptor agonist, THC, induces ER stress and inhibits the Akt/mTORC1 axis, which results in the activation of the p8/TRB3 pathway and induction of autophagy in human glioma cells [148]. Ultimately, cannabinoids are able to initiate autophagy and apoptosis via ER-stress-dependent mechanisms in concert with mitochondrial signaling.

## 6. Potential Impacts of Cannabinoids on Cellular Differentiation

As discussed earlier, cannabinoid activity at the levels of the ER and mitochondria may modulate specific intracellular signaling pathways that are crucial for maintaining stemness or initiating differentiation events. Among these pathways, the Wnt signaling pathway stands out due to its well-investigated role in stem cell homeostasis. In addition, cannabinoids may influence Wnt signaling through their interactions with PPARγ, as several studies have demonstrated a reciprocal negative regulation between Wnt-β-catenin and PPARγ pathways. Wnt signaling has long been established as a crucial signaling pathway involved in governing stemness, proliferation, and differentiation in adult mammalian tissue stem cells; however, there still remains a debate as to whether Wnt signaling activation promotes [31] or diminishes the self-renewal capacity of ESCs [30]. For instance, in the context of gastrointestinal tissue, intestinal stem cells abandon their self-renewal capacity and consequently differentiate into transit-amplifying (TA) cells following a Wnt-mediated signaling cascade. To elaborate, Wnt ligands bind and activate Frizzled receptors, a cytoskeletal adaptor protein and transcriptional co-regulator, β-catenin, accumulates and associates with Rac1 in order to translocate into the nucleus. Within the nucleus, β-catenin displaces the transcriptional repressor complex and subsequently binds to key transcription factors, such as TCF1, LEF1, and TCF4, to influence broader cellular processes including stemness, proliferation, and differentiation [149,150,151]. Interestingly, CB2 activation has been demonstrated to promote the nuclear translocation of β-catenin, which leads to exacerbated kidney fibrosis [152]. Similarly, in neurodegenerative diseases such as glaucoma and Alzheimer’s disease, Wnt signaling is downregulated, whereas GSK3-β, an inhibitor of the Wnt pathway, is upregulated [153]. Moreover, CBD has been demonstrated to downregulate GSK3-β, and, in turn, upregulate Wnt/β-catenin signaling in both models of neurodegeneration [154,155]. Additionally, CBD has been established as a PPARγ agonist, and previous studies have suggested that the inhibition of GSK3-β by CBD is mediated through PPARγ stimulation [154,156]. In human and mouse MSCs, CBD has been demonstrated to induce adipogenic differentiation by activating PPARγ, thus promoting lipid accumulation and the expression of adipogenic genes, effects that were reversed with the treatment of a PPARγ antagonist, T0070907 [157]. As Wnt signaling has been shown to inhibit adipogenesis [158], and the suppression of Wnt/β-catenin signaling by PPARγ has been established to promote adipocyte differentiation [159], this suggests that CBD-induced PPARγ activation may alter Wnt activation and dysregulate adipogenesis. As PPAR agonists have been implicated in inducing autophagy, CBD has also been found to induce autophagy by activating ERK1/2 and suppressing Akt in CB1-, CB2- and TRPV1-dependent manners in neural cells, a cellular process that maintains stem cell pluripotency [160]. Taken together, aberrant cannabinoid and ECS activity could dysregulate the cross-talk between key differentiation modulators, as highlighted in the context of adipogenesis, and may alter key cellular processes, such as autophagy, linked to stem cell self-renewal.

Moreover, by employing an in vitro model of mouse intestinal epithelial stem cells containing a mutant of β-catenin, LS174T, Heijmans et al. subjected cells to ER-stress-inducing conditions on the basis that intestinal stem cell differentiation is reliant on the induction of ER stress and UPR activation downstream in a PERK-eIF2α-dependent manner [161]. Based on previous studies, it is known that PERK-eIF2α activation arrests protein translation and results in the rapid depletion of nascent proteins. As such, a Wnt-dependent transcription factor, c-MYC, which is implicated in balancing self-renewal and differentiation aspects of stem cell function, is primarily affected by PERK-eIF2α arrests in protein translation due to having a short half-life [162]. Additionally, a proteomics screen revealed a novel transcription factor, CtBP2, whose expression was decreased following ER stress and rescued following PERK inhibition [163]. Given the observation that CtBP2 was overexpressed in mouse and human colorectal adenomas and that the inducible overexpression of CtBP2 partially restored stemness in ER-stress-induced LS174T cells, these findings suggest that CtBP2 is indeed implicated in regulating stemness [163] and that its expression may be regulated, in part, by the ER stress pathway.

## 7. Conclusions

We previously highlighted how endocannabinoids and phytocannabinoids can alter mitochondrial and ER function and signaling in the context of cellular differentiation and growth. First, the metabolic switch coordinated by mitochondria from being primarily glycolytic to oxidative is crucial in signaling stem cell commitment and differentiation [74,164]. Accordingly, this metabolic shift is accompanied by mitochondrial morphological changes, wherein mitochondria transition from being predominantly fragmented to being elongated [83,120]. Both AEA and 2-AG directly target mitochondria to diminish ATP production and membrane potential, whereas THC has been demonstrated to induce mitochondrial fragmentation [85], indicating that these CB1 receptor agonists may indirectly promote stemness by inhibiting the crucial metabolic switch required to exit pluripotency. Furthermore, cellular differentiation relies on sufficient ATP production to meet demanding energy requirements, which may be impaired in the presence of AEA and 2-AG. We discussed the necessity of autophagy in self-renewing and quiescent stem cells, and cannabinoids, such as CBD, have been shown to promote autophagy, thereby supporting pluripotency [160]. Indeed, at low micromolar concentrations, CBD increases cell proliferation in Jurkat cells, which could suggest concentration-dependent alterations to stem cell proliferation.

In breast cancer cells, AEA has demonstrated the ability to inhibit Wnt/β-catenin signaling. Following AEA treatments, there was a significant reduction in β-catenin levels and ultimately less transcriptional activation of the TCF responsive element, which is otherwise reflective of β-catenin signaling [165]. In contrast, a recent study identified increased expression of *Cnr1*^+^ hepatic progenitor cells (HPCs) undergoing liver regeneration along with increased AEA levels. They determined that AEA promoted nuclear localization of β-catenin and possessed the ability to modulate OXPHOS depending on substrate availability in HPCs [166]. Additionally, upon stimulation of CB1 (encoded by *Cnr1*) with 50nM AEA, they observed increased HPC proliferation [166], suggesting a potential avenue for CB1 agonists to modulate the proliferative capacity of stem cells. Furthermore, Nalli et al. have demonstrated that CBD and its homolog, cannabidivarin, are able to inhibit Wnt/β-catenin signaling in a dose-dependent manner [167]. Ultimately, the inhibition of Wnt/β-catenin signaling could pose implications for cellular differentiation in that β-catenin would otherwise be required to upregulate the transcription of stem cell identity-affiliated genes. Alternatively, with regards to ER stress-induced stem cell differentiation, (endo)cannabinoids including AEA, 2-AG, and THC have been shown to induce ER stress and activate the UPR, thus promoting the acute depletion of CtBP2 and the initiation of cellular differentiation in a PERK-eIF2 dependent manner [144,145,148]. All in all, due to the integrity of the ECS across several tissue systems, there exist many potential routes through which cannabinoid-induced dysregulation of cellular differentiation may occur; however, this idea requires further investigation to elucidate whether cannabinoids impact stem cell homeostasis. Metabolic pathways are differentially regulated in order to mediate stem cell fate decisions, and any perturbances could result in stem cell dysfunction and ultimately pathological consequences. We discussed earlier how mTORC1 and mTORC2 function in opposition to regulate cellular differentiation, and taking into consideration the recent evidence that THC inhibits the Akt/mTORC1 axis [148], we presume a potential point of metabolic dysregulation resulting from phytocannabinoid activity that results in stunted cellular differentiation. Taken together, it will be critical to consider the broad implications of disruption of metabolic homeostasis by cannabinoids in trophoblast cells, including immune signaling and oncogenesis, and its associated therapies.

The ability of stem cells to retain their key functional characteristics, in other words, to be able to selectively maintain stemness or commit and differentiate into specific sub-lineages, is a crucial process that renders developmental and/or regenerative potential to certain tissues. Embryonic differentiation and placentation are key events contributing to early fetal development and relying on the functions of embryonic stem cells and trophoblast stem cells, respectively. Ultimately, cannabis use during gestation has been linked to poor maternal and fetal outcomes, including pre-eclampsia, low fetal birthweight, preterm birth, and stillbirths. As such, understanding the impacts of cannabinoids on stem cell homeostasis will be critical in order to avoid disruption of fetal development in the event that these unwanted pregnancy outcomes result from cannabinoid actions on these stem cell populations. Similarly, adult stem cells, such as those found in the liver, bone, brain, and skeletal muscle tissues, are responsible for regeneration following tissue injury or death. Without their inherent self-renewal capacity, the adult stem cell pool would become exhausted and thus lack the long-term regenerative potential that is necessary to sustain recurring tissue injuries throughout life. Thus, despite the accumulating evidence suggesting that cannabinoids impact stem cell function, there remain gaps in the literature that need to be urgently investigated.

## Figures and Tables

**Figure 1 ijms-24-11003-f001:**
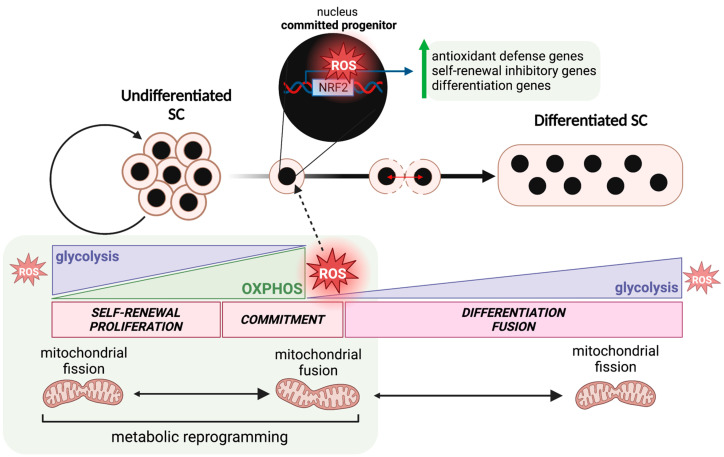
Interplay of mitochondrial dynamics and metabolic reprogramming during stem cell differentiation. Stem cells either undergo self-renewal in order to maintain the stem cell population or undergo differentiation, giving rise to more specialized cell types. Self-renewing and quiescent stem cells, such as hematopoietic stem cells (HSCs), neural stem cells (NSCs), and mesenchymal stem cells (MSCs), possess low metabolic activity, generally favoring glycolytic metabolism, and exhibit fragmented mitochondrial phenotypes. The metabolic shift from glycolysis to oxidative phosphorylation (OXPHOS) marks the onset of stem cell commitment and differentiation, which is accompanied by a transiently elongated mitochondrial phenotype and greater reactive oxygen species (ROS) production. In NSCs, acute ROS production serves to activate downstream nuclear targets, thereby upregulating NRF2 transcription and promoting NSC differentiation gene expression. Created with Bio Render.com. Accessed on 19 June 2023.

**Figure 2 ijms-24-11003-f002:**
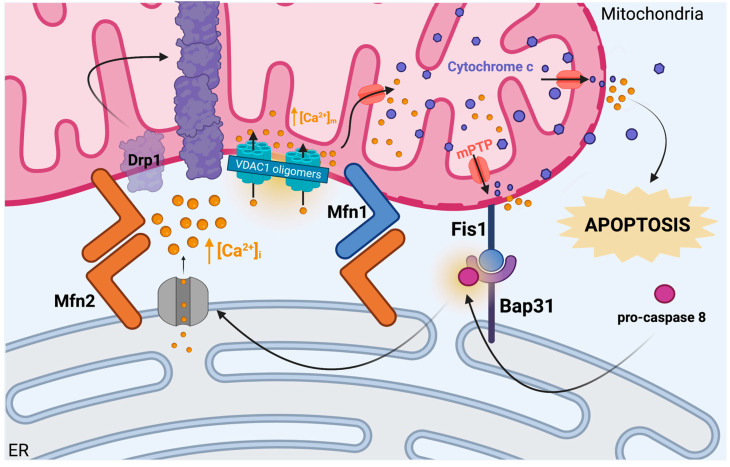
Mitochondria-ER cross-talk in calcium sensing and apoptotic regulation. Mitochondria-associated membranes (MAMs), including Mfn1-Mfn2 dimers and Fis1-Bap31 binding sites, join mitochondrial and ER membranes and serve to create microenvironments and proximity for cross-talk between the organelles. An early event in apoptosis induction occurs when pro-caspase 8 is recruited to and activated at Fis1-Bap31 sites, where it cleaves Bap31 and ultimately promotes the release of Ca^2+^ stores from the ER lumen into the mitochondrial-ER microenvironment. In response to mitochondrial Ca^2+^-overload, mPTP forms, and VDAC1 oligomerization renders the mitochondrial membrane permeable to small molecules. Drp1 translocation and activation are stimulated shortly thereafter, triggering the release of cytochrome *c* through mPTP and VDAC1, leading to apoptosis. The arrows indicate the coupling of procaspase 8 to Bap31 and the subsequent influence of this action on Ca^2+^ release from the ER. Created with Bio Render.com. Accessed on 23 June 2023.

**Figure 3 ijms-24-11003-f003:**
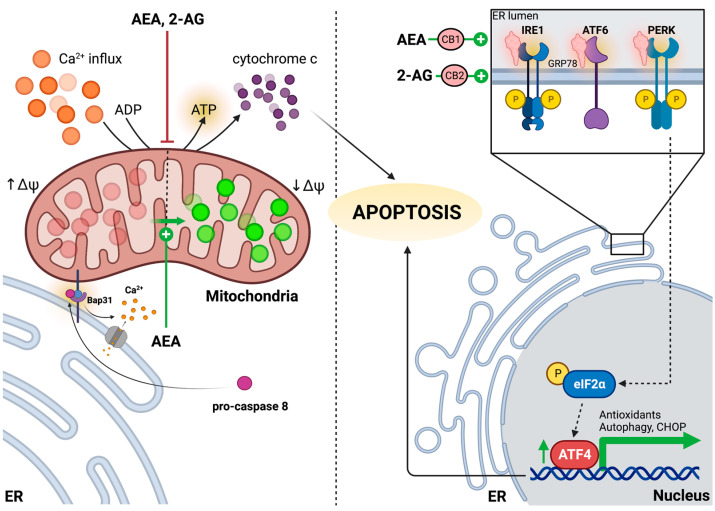
Endocannabinoid dysregulation of mitochondria and ER signaling. AEA and 2-AG block the production of ATP and prevent the Ca^2+^ influx into mitochondria, which may have been dispensed from the ER lumen following Fis1-Bap31 stimulation by pro-caspase 8. AEA promotes mitochondrial membrane depolarization (decreased J-aggregates in red; increased J-monomers in green), thus lowering the electrochemical driving force required for ATP production and priming the mitochondrion for apoptosis. 2-AG promotes the upregulation and localization of GRP78 to UPR-associated receptors, including IRE1, ATF6, and PERK, and promotes the upregulation of ATF4, a transcription factor that promotes the transcription of CHOP and genes related to autophagy and antioxidant production. AEA activates all major branches of the UPR and also upregulates CHOP in a PERK-mediated manner. AEA and 2-AG have been postulated to exert their ER-stress-mediated effects through CB1 and CB2, respectively. Left panel: arrows indicate either mitochondrial functions or procaspase 8 activation and its downstream effects associated with apoptosis induction. Right panel: arrows indicate downstream transcriptional upregulation of ATF4 following UPR activation by AEA and 2-AG, which lead to apoptosis. Created with Bio Render.com. Accessed on 14 May 2023.

## Data Availability

Not applicable.

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
