# Peer review of "The Implications of Cannabinoid-Induced Metabolic Dysregulation for Cellular Differentiation and Growth"

_ijms, 2023, doi:10.3390/ijms241311003_

Round 1

Reviewer 1 Report

 In this work Podinić and colleagues explore the implications of the metabolic changes induced by cannabinoids in the cell differentiation and growth. The work is presented in a well organized format,  and the topic is relevant in many aspects, however some issues must be attended:

Some metabolic changes promoted by cannabinoids and important to cell fate are not considered within this article e.g., the effect of cannabinoids on lipogenesis or autophagy (e.g., https://doi.org/10.1016/j.bcp.2022.114910 , DOI:https://doi.org/10.1038/s41385-021-00455-x).  the authors may want to improve their manuscript content by discussing these important effects.

L36 canabinoids regulate other members of the TRP family and also other GPR receptors (e.g. GPR18), the authors may want to consider a general mention to this families (TRP and orphan GPCRs).

3.3/ L280: indeed, cannabinoids directly interact with mitochondria as evidenced in experiments with isolated mitochondria doi: 10.1038/s41419-019-2024-0. As discussed by authors, CBD regulates the mitochondrial Ca2+ homeostasis, favoring the proliferation at low concentrations (0-5 mM) and promoting cell death by mitochondrial swelling when employing concentrations superior to 10 mM, as you know, Ca2+ fuels many of the TCA enzymes to promote metabolism, however when signals are persistent of excessive mPTP opening is promoted. The authors may want to discuss that the mitochondrial Ca2+ regulation by cannabinoids is complex and may depend on the concentrations employed.

L419. I respectfully do not share the appreciation of an overwhelming release of calcium ions from ER. In their work the authors employed an intensiometric dye to estimate differences (not concentrations as they mentioned) in cell populations MFI by flow cytometry, which is an unpopular method to evaluate intracellular calcium for several weaknesses. Despite the authors interpretation, the cytosolic basal levels only changed from 100% to 120-140%? Considering the resting basal levels of most of the cells (near to 100 nM) the change generated by Fis expression if very limited.  As you know Ca2+ can move to the intermembrane space using VDAC, however the MCU activity does depends on considerable calcium increases to allow its opening (in a micromolar levels). Additionally, as far as I know, the role of Drp1 in cytochrome c release is totally dispensable and not significant in comparison with other mechanisms. Some authors even discuss a protective role of Drp1 against apoptosis (DOI: 10.1016/j.molcel.2004.09.026; DOI: 10.1111/bph.12515).  

The idea of Figure 1 is simpler. ER-mitochondrial contacts are determinant for cell fate in 2 ways, if controlled they “feed” mitochondrial metabolism, but when deregulated the excessive ER release favors MCU opening (VDAC does not represent a barrier for calcium)  and promotes mPTP recruitment.  Which conformation is indeed regulated for excessive calcium and ROS (the latter a consequence of Ca2+ overload). mPTP formation mediates the liberation of cytochrome c…  Mitochondrial fission (Drp1 recruitment) is usually activated as a way to facilitate the clearance of this damaged mitochondria (mitophagy)…   the authors may want to consider this idea.

L497: I think that the idea is not correct. “Work by Catanzaro et al. suggests that AEA operates in a dose- dependent manner to increase mitochondrial swelling and simultaneously desensitize mitochondria to intracellular Ca2+ levels by decreasing cytochrome c release in response to calcium “.  The author’s results do not show that AEA increases the mitochondrial swelling.  By the contrary, AEA only reduced but did not avoid neither the calcium mediated swelling nor the cytochrome release and they only mention a dose-dependent effect, but it is not shown.

L500 “Likewise, 2-AG inhibits cyclosporin A sensitive Ca2+-dependent cyto- 500 chrome c release in rat liver mitochondria.” the sentence is confusing for me, so I decided to see the original research, but the reference is missing….

Figure 3. I think that the information is sometimes misinterpreted and is giving a false message e.g. L506 “AEA and 2-AG block 506 the production of ATP and prevent the Ca2+ influx into mitochondria” there is no evidence for this statement.    Additionally I found that Figure 3 is very confusing, the reader may understand that cytochrome C release is a normal process which is not. ATP production and cytochrome C release is regulated by calcium by very complex mechanisms that are far from been explain by this figure.  Moreover, the meaning of the red and green circles in the matrix must be stated. If they represent the proton gradient, they have to be placed in correct place.

Minor comments

There is a problem with the Figures and corresponding figure legends, they are randomly inserted in the text.

Figure 1: based on its information, one may understand that during commitment either progenitor or differentiated cells loss their mitochondria

L295 Calcium must be defined here instead of L349

L297 respectfully> respectively

L354, L360, L361… and so on…. calcium > Ca2+

L361 mitochondrial Calcium overload, particularly

In vitro, et al., must be in italics

L364 calesequestrin > calsequestrin

Author Response

June 23, 2023

Responses to reviewers for  Manuscript ijms-2432675

Title: The Implications of Cannabinoid-Induced Metabolic Dysregulation for

Cellular Differentiation and Growth

We thank all four reviewers for their time and diligent analysis of the manuscript as well as their helpful suggestions. While all these suggestions are pertinent to the topic, a wholesome discussion of all of these suggestions would substantially lengthen this review. We have tried to highlight the important messages suggested by the reviewers as the focus of our revisions and trust the reviewers will find this to be acceptable.  Below is a detailed accounting of how we have addressed the individual concerns of the reviewers.

R1 comment L36: Cannabinoids regulate other members of the TRP family of receptors as well as other G-protein receptros, please include a general mention.

Lines 36-38 have been modified to include this additional information.

R1: Comment L280: The authors may want to discuss that Ca2+ regulation by cannabinoids is complex may depend on the concentrations employed

We thank the reviewer for highlighting this issue. While a detailed discussion of this complex topic is beyond the scope of the current review, it is an important area for consideration in the context of mitochondrial health. Therefore, we have inserted a few sentences underscoring the reviewer’s point between lines 319-328.

R1 comment, L419: The reviewer does not agree of with the observations reported in the citated paper that regarding the possibility of an overwhelming release of Ca2+ from the ER and the possible role in the recruitment of DRP-1.  

We thank the reviewer for this important perspective and made note of the possible controversy around this point in line 497-498 and added the suggested citation.

R1 comment, L497: I think that the idea is not correct. "Work by Catanzaro et al. suggests that AEA operates in a dose- dependent manner to increase mitochondrial swelling and simultaneously desensitize mitochondria to intracellular Ca2+ levels by decreasing cytochrome c release in response to calcium". The author's results do not show that AEA increases the mitochondrial swelling. By the contrary, AEA only reduced but did not avoid neither the calcium mediated swelling nor the cytochrome release and they only mention a dose-dependent effect, but it is not shown.

We thank the reviewer for their perspective and for stimulating us to more deeply consider the cited paper and we offer the following rationale for current citation. According to the literature, the measurement of mitochondrial swelling by DA540 values is an established, accepted technique in the literature [1]. To highlight an excerpt from Javadov et al: “The main method widely used for estimation of the mitochondrial volume is based on the measurement of absorbance (optical depth) in isolated mitochondria. The amount of light scattered by mitochondria depends on the matrix volume, where a decrease of absorbance is proportional to an increase of mitochondrial volume (Tedeschi and Harris, 1958).” The reviewer commented that a study by Catanzaro et al. cited in our paper failed to show that AEA increases mitochondrial swelling. In the review paper above, the author emphasizes that measuring the decrease in A540 absorbance to examine mitochondrial swelling is a common technique used in studies with isolated mitochondria, and the study by Catanzaro et al. uses isolated mice liver mitochondria. In Figure 1 they show that 50µM of AEA leads to a time-dependent maximal decrease in A540 absorbance of 42%(Catanzaro et al. 2009), which corresponds to increased mitochondrial swelling and supports the notion originally described in our review paper. Furthermore, the paper shows that 50µM AEA treatment alone significantly increases mitochondrial swelling, without altering cytochrome c release, whereas 100µM Ca2+ treatment significantly increased cytochrome c release and mitochondrial swelling (respective to control). Interestingly, pre-treatment of isolated mitochondria with 50µM AEA before the addition of 100µM Ca2+ diminished the Ca2+-induced cytochrome c release by 50%, to which the authors conclude that AEA is able to reduce the sensitivity of mitochondria to Ca2+ ions.

R1 comment L500: A missing reference for the statements in line 500 regarding the effects of 2-AG on Ca2+ and cytochrome c.

We thank the reviewer to identifying this and we have inserted reference #139 (Zaccagnino, P., et al., The endocannabinoid 2-arachidonoylglicerol decreases calcium induced cytochrome c release from liver mitochondria. J Bioenerg Biomembr, 2012. 44(2): p. 273-80).

R1 comment Figure 3 is confusing.  A reader may understand that cytochrome c release is a normal process, it is not.

Thank you for sharing this perspective. We have replaced Figure 3 with one which hope will be clearer. We have indicated the process which are impacted by endocannabinoids.

R1 Minor comments

  • Figure 1 has been clarified as per the reviewers suggestion.
  • Respectfully has been changed to respectively at L297
  • Calcium has been defined at first occurrence.

R2 Comment: Figure legends should be listed below the figures

Our apologies. The legends may have shifted during final formatting. We thank the reviewer for noticing this and we have corrected the formatting. Please note that Figure 3 as well as its associated legend has been replaced.

R2 comment: In Line 142-169, as reveled, Cannabinoid may affect AMPK- mTOR to affect tumor cell growth. But, due to the efficacy of AMPK-mTOR in affecting PD-L1 expression, the possible effects of Cannabinoid in affecting tumor therapy through affecting immune sensitivity should be added. Some references should be added to this part including 10.1016/j. jconrel.2022.11.004

We thank the reviewer for pointing to the importance of the connection between metabolic dysregulation and cancer therapy.  This is vast area of research and we have not addressed this in current manuscript because this may draw the discussion away from the mechanistic focus of the paper between cannabinoids, mitochondria and cellular differentiation.  While there are admittedly overlaps, a respectful discussion of the area would necessitate a substantial increase in the length of the manuscript and obfuscate the central messages of the current review.

R2 comment: How some other immune checkpoints was affected by Cannabinoid in solid tumors should be more clearly reviewed in this review if possible, including CTLA-4, VISTA, LAG-3, TIGIT and TIM-3.

We thank the reviewer for pointing to importance of the dysregulation of immune checkpoints by cannabinoids. However, the scope of our review is mainly focused on the dysregulation of stem cell function through metabolic changes induced by cannabinoids, and the inclusion of these immune checkpoints in our discussion would need to be contextualized in framework of trophoblast immune function. This would substantially expand the scope of the review.

R2 comment: How does cannabinoid-induced metabolic dysregulation affect the sensitivity of some other tumor therapies could be added including chemotherapy, radiotherapy,and CRT.

This is indeed an important area of investigation and, we thank the reviewer for pointing to an important future direction. However, this interrelationship is complex and very extensive due the number of therapeutic strategies touched upon by the reviewer. However, to acknowledge the importance of the reviewers observations, we have inserted a summative statement in Line 770-773.

R3 comment: The authors could more clearly state their research question

We thank the reviewer for pointing out this possible confusion. Since this is a narrative review we have stated the focus of this review clearly in the last sentence of the abstract rather stating a specific hypothesis or research question.

R3, R4 comments: General Methodology, Quality Assessment and Data Synthesis Statement

We employed a narrative review design for “The Implications of Cannabinoid-Induced Metabolic Dysregulation for Cellular Differentiation and Growth” in attempt to (1) summarize the literature pertaining to cannabinoids and cellular function and (2) extrapolate the impacts of cannabinoids on stem cell fate decisions by examining the effects of cannabinoids on key cellular functions, such as mitochondrial metabolism and ER dynamics, which are positioned as central regulators of stem cell fate decisions. As a baseline, all of the studies incorporated in this review are peer-reviewed, English language, and are not restricted in-terms of publication date. We created a list of some keywords that were imperative in conducting the literature search, including: stem cells; cannabinoids; mitochondria; cellular differentiation; endoplasmic reticulum; oxidative stress; cellular metabolism. When inconsistencies or a lack of consensus was encountered during the literature search, we continued to provide both arguments, with the caveat that there is “debate” or “controversy” surrounding the exact mechanism being investigated. It is extremely important to us to convey both sides of controversial scientific topics, in order to keep the reader informed and aware that this paper is not making finite claims. As mentioned above, our primary focus centers on how cannabinoids could dysregulate stem cell functions, based on the existing body of literature.  Generally, this type of narrative review does not require the incorporation of the methodology into the body of the manuscript.

R3 comment: The authors should provide more details and greater clarification.

We thank the reviewer for this perspective.  It is important to ensure that the manuscript is clear. Based on the on the comments of all the reviewers we have included a few more examples, altered some of the figures and all the while attempting to respectfully maintain the length of the manuscript so it remains focused. We hope the additional modifications have addressed the concerns of this reviewer.

R4 comment:Overall the research question is clear and well-grounded in the existing literature, providing more specific details about the hypothesis and objectives can further enhance its clarity.”   “ The authors have also done a commendable job in discussing a wide range of studies, including those that investigate the effects of different cannabinoid's, use different cell types, and explore different aspects of mitochondrial and ER function”

We thank the reviewer for these kind comments. We have added additional text and modified our figures to help with the clarity of the manuscript. It is necessary to include some technical language in order to convey the important signaling elements impacted by cannabinoids. However, we have attempted to consolidate the interaction of these pathways visually through our figures. We hope this facilitates the understanding of the reader who is less well versed in signaling pathways but still impacts an appreciation for central focus of the review.

Kind Regards,

Sandeep Raha

[email protected]

Reviewer 2 Report

In this research, the authors reviewed the recent development of The Implications of Cannabinoid-Induced Metabolic Dysregulation for Cellular Differentiation and Growth. Generally, it’s meaningful and interesting review. In my opinion, the current version of this manuscript fits the scope of International Journal of Molecular Sciences and could be accepted after major revision.

My specific comments are in detail listed below:

1.     The figure legends (Figure 1-2) should be listed below the figures. The authors should check it.

2.     The role of AMPK-mTOR in tumor cell differentiation and growth should more clearly discussed in the Cannabinoid-induced metabolic dysregulation for cellular differentiation and growth. Some references could be added including 10.1016/j.ijbiomac.2022.10.167.

3.     The figure legend of Figure 3 was lost?

4.     How some other immune checkpoints was affected by Cannabinoid in solid tumors should be more clearly reviewed in this review if possible, including CTLA-4, VISTA, LAG-3, TIGIT and TIM-3.

5.     In Line 142-169, as reveled, Cannabinoid may affect AMPK-mTOR to affect tumor cell growth. But, due to the efficacy of AMPK-mTOR in affecting PD-L1 expression, the possible effects of Cannabinoid in affecting tumor therapy through affecting immune sensitivity should be added. Some references should be added to this part including 10.1016/j.jconrel.2022.11.004.

6.     Some minor mistakes existed in this paper. The authors should carefully check it.

7.     How Cannabinoid-induced metabolic dysregulation affect the sensitivity of some other tumor therapies could be added including chemotherapy, radiotherapy, and CRT.

Author Response

(The authors gave the same response as above.)

Reviewer 3 Report

Below are my specific observations and recommendations for enhancement.

1. Research Question: The authors have presented a comprehensive review on the role of endocannabinoids and phytocannabinoids in mitochondrial and endoplasmic reticulum (ER) function. The topic is timely and significant, considering the growing interest in the therapeutic potential of cannabinoids. However, the research question could be more explicit. For instance, the authors could state their hypotheses or specific research objectives more clearly.

2. Methodology: The authors have used a comprehensive literature review approach, which is a valid method for this type of study. However, the methodology lacks details on inclusion and exclusion criteria, quality assessment, and data synthesis. These improvements could enhance the transparency and robustness of their methodology.

3. Clarity and Structure: The paper is logically structured, but it contains dense technical language that could be simplified for a wider audience. The flow of the paper could be improved with more linking sentences and a clearer summary of the main findings.

4. Discussion: The authors have provided a detailed discussion on the role of endocannabinoids in mitochondrial and ER function. However, the discussion could be enhanced by providing more specific examples and delving deeper into the mechanisms and implications of these effects.

5. Implications: The authors have highlighted the potential implications of their findings on cellular differentiation and growth. However, they acknowledge the need for more studies to fill the existing gaps in knowledge.

Overall, this paper provides valuable insights into the impact of endocannabinoids on mitochondrial and ER function. However, there are areas that could be improved to enhance the clarity and robustness of the research.

Author Response

(The authors gave the same response as above.)

Reviewer 4 Report

Please find below my specific comments and suggestions for improvement.

1.    The research question is centered around the role of endocannabinoids and phytocannabinoids in mitochondrial and endoplasmic reticulum (ER) function, and their potential impact on cellular differentiation and growth. This is a significant and timely topic given the increasing interest in the therapeutic potential of cannabinoids and the crucial roles of mitochondria and ER in cellular function and survival. The authors have done a commendable job in clearly stating this research question and purpose. They have provided a comprehensive background of the current understanding of the endocannabinoid system, mitochondrial and ER biology, and the potential interactions between these systems. This sets a solid foundation for their research and allows readers to understand the context and significance of their work. However, the authors could further improve the clarity of the research question by explicitly stating their hypotheses or specific research objectives. For instance, are they hypothesizing that specific cannabinoids have differential effects on mitochondrial and ER function? Are they expecting these effects to be cell type-specific or universally applicable across different cell types? Are they exploring the potential therapeutic or detrimental implications of these interactions? Explicitly stating these details can help readers better understand the scope and direction of the study. Overall, while the research question is clear and well-grounded in existing literature, providing more specific details about the hypotheses or objectives can further enhance its clarity.

2.    The authors have used a comprehensive literature review approach to gather and analyze existing data on the topic. This is a valid method for synthesizing current knowledge and identifying gaps or inconsistencies in the literature. The authors have also done a commendable job in discussing a wide range of studies, including those that investigate the effects of different cannabinoids, use different cell types, and explore different aspects of mitochondrial and ER function.

However, the methodology could be improved in several ways:

Inclusion and exclusion criteria: The authors should clarify the criteria they used to include or exclude studies in their review. This can help readers understand the scope of the literature review and assess the potential for selection bias.

Quality assessment: The authors should consider conducting a quality assessment of the included studies. This can help readers understand the strength of the evidence supporting the authors' conclusions.

Data synthesis: The authors could provide more details about how they synthesized the data from the included studies. Did they use any specific frameworks or tools for data synthesis? How did they handle inconsistencies or contradictions in the data?

Overall, while the authors' methodology is sound in general, providing more details about the literature search strategy, inclusion and exclusion criteria, quality assessment, and data synthesis can enhance the transparency and robustness of their methodology.

3.    The authors have done a commendable job in structuring the paper in a logical and coherent manner. The paper is divided into clear sections that guide the reader through the background of the topic, the main findings from the literature, and the authors' conclusions and future directions. This structure helps the reader understand the flow of the paper and the authors' line of reasoning.

However, there are several areas where the clarity and structure could be improved:

Technical language: The paper is quite dense and uses a lot of technical language and jargon. While this is appropriate for a scientific audience, the authors could do a better job of explaining some of the more complex concepts and terms. This would make the paper more accessible to a wider audience, including scientists who are not experts in this specific field.

Linking sentences: The authors could improve the flow of the paper by using more linking sentences at the beginning and end of each section. These sentences can help the reader understand how each section relates to the next and how they all fit together to support the authors' overall argument.

Summary and implications: The authors could do a better job of summarizing their main findings and discussing their implications. This would help the reader understand the significance of the paper and how it contributes to the field.

Overall, while the clarity and structure of the paper are generally good, the authors could improve the paper by explaining complex terms more clearly, using more visual aids, improving the flow with linking sentences, and providing a clearer summary and discussion of implications.

4.    The authors have done a commendable job in discussing the role of endocannabinoids in mitochondrial and ER function and signaling. However, the discussion could be improved by providing more specific examples of how these effects are observed in different cell types or under different physiological conditions. For instance, the authors could discuss how endocannabinoid signaling affects mitochondrial function in neurons versus muscle cells, or how these effects might change under conditions of stress or disease. Additionally, the authors could delve deeper into the mechanisms by which endocannabinoids affect mitochondrial function. While the authors mention that endocannabinoids can diminish ATP production and membrane potential, they could provide more detail on how these effects are mediated at the molecular level. Furthermore, the authors could also discuss more about the implications of these effects on cellular function and health. For example, how might changes in mitochondrial function due to endocannabinoid signaling affect cell survival, proliferation, or differentiation? And how might these effects contribute to the development or progression of diseases? Lastly, while the authors do mention some of the key players in endocannabinoid signaling (such as AEA and 2-AG), they could provide a more comprehensive overview of the endocannabinoid system, including the different types of endocannabinoids and their receptors, and how these components interact to regulate cellular function.

5.    The authors discuss the potential implications of their findings on the understanding of cannabinoids' effects on cellular differentiation and growth. They emphasize the need for further research to fully comprehend the impact of cannabinoids on stem cell function and the consequences for fetal development and tissue regeneration. The authors also highlight the potential risks of cannabis use during pregnancy on fetal health. However, they acknowledge the need for more studies to fill the existing gaps in knowledge.

Author Response

(The authors gave the same response as above.)

Round 2

Reviewer 2 Report

The current version of this manuscript could be accepted.